# Fast Learning in Balanced Deep Spiking Neural Networks with Strong and Weak Synapses

## Abstract

The intricate neural dynamics of the cerebral cortex are often characterized in terms of the delicate balance between excitation and inhibition (E-I balance). While numerous studies have delved into its functional implications, one fundamental issue has remained unresolved – namely, *the unstructured, random connections posed by E-I balance dynamics versus the necessity for structured neural connections to fulfill specific computational tasks*. This raises the crucial question: How can neural circuits reconcile these seemingly contradictory demands? Drawing inspirations from recent data in neuroscience, we propose a biologically grounded spiking neural network. This network incorporates two distinct sets of synaptic connections, one featuring strong synapses dedicated to maintaining the balance condition, and the other comprising weak synapses utilized for neural computation. Crucially, only the weak synapses undergo training, while the strong synapses remain fixed. Interestingly, we have discovered that this architecture not only resolves the structural conflicts, but also offers several compelling computational advantages. Firstly, the E-I balance dynamics mediated by strong synapses can closely mimic the function of normalization operations, effectively alleviating the internal covariate shift problem. Secondly, we have observed that weak synapses remain weak during training without any imposed constraints, thus preserving the balance condition established by the strong synapses. Lastly, the coexistence of strong and weak synapses allows for a seamless transition from the "lazy" learning regime, characterized by the primary training of readout weights, to the "rich" learning regime, marked by alterations in neural representations. We believe this study can shed light on how structured computations can coexist with unstructured E-I balance dynamics and offer novel perspectives on the computational advantages of E-I balance.

## 1 Introduction

The concept of excitation-inhibition balance (E-I balance), initially introduced to elucidate the irregular firing patterns observed in the cortex (Softky & Koch, 1993; Shadlen & Newsome, 1994), has emerged as a foundational principle in neuroscience. The E-I balance model posits that, in a randomly connected networks with strong weights, neurons receive a constant influx of excitatory postsynaptic currents (EPSCs) and inhibitory postsynaptic currents (IPSCs), resulting in membrane potentials exhibiting a random-walk-like behavior and, consequently, generating irregular spike trains (van Vreeswijk & Sompolinsky, 1996; van Vreeswijk & Sompolinsky, 1998). Empirical investigations consistently affirm the equilibrium maintained between excitatory and inhibitory inputs to neurons (Shu et al., 2003; Wehr & Zador, 2003; Haider et al., 2006; Okun & Lampl, 2008; Xue et al., 2014; Barral & D Reyes, 2016), with deviations from this balance associated with various neural disorders (Yizhar et al., 2011).

The computational advantages bestowed by maintaining E-I balance in cortical dynamics are multifaceted and profound. In this balanced state, neurons exhibit heightened responsiveness to incoming stimuli and can swiftly process such inputs van Vreeswijk & Sompolinsky (1998); Huang et al. (2011); Tian et al. (2020). Furthermore, this equilibrium condition affords dynamic range control, enabling the neural system to handle a broad spectrum of inputs, ranging from subtle sensory signals to salient events. Additionally, globally coordinated inhibition in balanced dynamics can yield optimal coding, even when the firing of signal neurons adheres to Poisson statistics (Boerlin et al.,

2013; Denève & Machens, 2016). Lastly, the ability to adjust the E-I ratio effectively acts as a gate for signals (Vogels & Abbott, 2009; Kremkow et al., 2010).

However, E-I balance dynamics do not operate in isolation. The brain also necessitates structured connections to fulfil a multitude of cognitive functions. A fundamental issue arises from the inherent conflict between E-I balance dynamics, which require unstructured connections, and the neural computation dynamics, which require structured connections. *How, then, does the brain resolve this inherent conflict?*

In this study, we introduce a biologically grounded spiking neural network, featuring distinct sets of strong and weak synapses. Our model draws inspiration from recent neuroscience data where Scholl et al. (2021) illuminated the pivotal role of synaptic strength in shaping the response of neocortical neurons to sensory input. Interestingly, this research challenges conventional views by revealing that individual neuron selectivity is not dictated by strong synapses but instead hinges on the collective activation of weaker synapses. This intriguing discovery aligns with the paradox where classical E-I balance network models scale with $1/\sqrt{N}$ (van Vreeswijk & Sompolinsky, 1998) ($N$ denotes network size), while empirically neural computation should scale with $1/N$ to maintain input current consistency across varying network sizes. These disparities form the basis of our hypothesis: the brain employs structured and weak synapses (synapses that scale with $1/N$) for neural computation while utilizing randomized and strong synapses (synapses that scale with $1/\sqrt{N}$) to maintain balanced dynamics.

The presented model comprises three distinct neuronal populations at each layer (Fig. 1A). Notably, our model incorporates two sets of synapses. One set consists of strong synapses responsible for maintaining the balance condition, while the other set comprises weaker synapses designated for neural computations (Fig. 1B). Consequently, excitatory neurons receive large input currents from the E-I balance dynamics and small input currents from an optimizable dynamics (Fig. 1C). Importantly, only the weaker synapses undergo training, while the stronger synapses remain fixed throughout the learning process. Our results reveal that this architecture not only resolves the aforementioned conflict in the structural demands, but also offers several compelling computational benefits. First, the E-I balance dynamics function computationally similar to normalization operations. Maintaining a balanced state stabilizes the distribution of network variables and thus expedite training. Intriguingly, we observed that the weak synapses, even without constraints, remain weak during training, ensuring the network remains balanced throughout. Furthermore, our study showcases that the combination of strong and weak synapses facilitates a smooth transition from a "lazy" learning regime, primarily training readout weights, to a "rich" learning regime, where neural representations undergo alteration. To empirically verify the computational merits of the weak-and-strong synaptic architecture, we conducted experiments utilizing the MNIST and Fashion-MNIST datasets, yielding promising results. Our findings provide insights into the diverse functions of synapses in the brain and shed light on how the brain leverages the balanced state to enhance its learning processes.

## 2 THE NETWORK MODEL

### 2.1 BIOLOGICAL SIGNIFICANCE

In deep neural networks, neurons can form both positive and negative connections with downstream neurons. However, the brain takes a different approach, categorizing neurons into either excitatory or inhibitory types. Excitatory neurons exclusively form positive connections with downstream neurons, while inhibitory neurons exclusively form negative connections with downstream neurons. This principle is widely known as Dale's principle within the neuroscience community. To align with this biological concept, our model consists of three distinct neuronal populations. The excitatory population corresponds to pyramidal cells (labeled as $E$), one of the principal excitatory neuron types in the brain. The inhibitory neurons correspond to somatostatin (SST)-expressing neurons (labeled as $I_d$) and parvalbumin (PV)-expressing neurons (labeled as $I_p$). SST neurons predominantly offer distal inhibition to pyramidal cells (Gonchar, 1997; Yavorska & Wehr, 2016), resulting in less effective inhibition, whereas PV neurons primarily provide on-path proximal inhibition to pyramidal cells, yielding strong inhibition (Packer & Yuste, 2011; Hu et al., 2014).

Crucially, we hypothesize that the interactions between the excitatory group and the two inhibitory groups serve different computational purposes. Specifically, the neural dynamics between the ex-

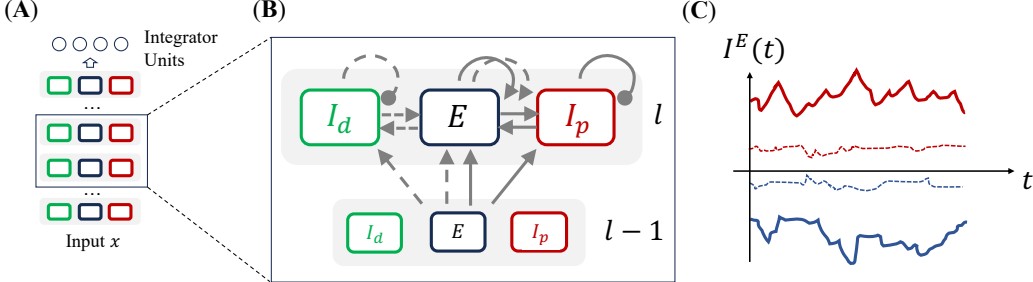

Figure 1: A balanced spiking neural network satisfying Dale's principle, featuring both strong and weak synapses. (**A**) The network architecture. We use non-leaky integrator neurons as readout units. (**B**) In each layer, our model comprises an excitatory population and two inhibitory populations. Please refer to Sec.2.1 for the biological significance. Each layer incorporates two sets of synapses, with only the weak set (dashed line) subject to training. Arrow and filled circle represent excitatory and inhibitory synapses respectively. (**C**) A schematic depiction of input currents to an excitatory neuron. The neuron recieves a barrage of large EPSCs and IPSCs mediated by strong synapses from the E-I balance dynamics, along with small ones mediated by weak synapses from an optimizable dynamics. Red and blue lines represent excitatory and inhibitory currents, respectively. Solid and dashed lines represent currents mediated by strong and weak synapses, respectively.

citatory and SST neurons involve weaker synapses and can be fine-tuned to perform specific tasks. On the other hand, the neural dynamics between the excitatory and PV neurons feature globally distributed, unstructured, and robust synapses (Packer & Yuste, 2011), which help maintain the network's dynamics within the balanced region. This configuration aligns with the different levels of inhibition provided by SST and PV neurons and is consistent with recent experiments where PV neurons are found to have critical roles in E-I balance dynamics (Xue et al., 2014; Ferguson & Gao, 2018), whereas the contribution of SST neurons in this context remains less evident. Lastly, the distinct functions of weak and strong synapses within the excitatory group are in accordance with recent findings in (Scholl et al., 2021), where strong synapses exhibit no correlation with neuronal selectivity, while the co-activation of weak synapses determines selectivity.

## 2.2 MODEL DETAIL

We use leaky integrate-and-fire (LIF) neuronal dynamics, which is given by,

$$\tau_m^b \frac{dV_{i,l}^b}{dt} = -g_L V_{i,l}^b + I_{i,l}^b, \tag{1}$$

where $b = E, I_p, I_d$ denotes the neuron type. $l$ denotes the layer index, and $i$ denotes the neuron index in that layer. $\tau_m^b$ is the membrane time constant, $g_L$ is the leaky conductance, and $I_{i,l}^b$ is the synaptic current received by the neuron.

For simplicity, we only present the synaptic current received by an $E$ neuron (Fig. 1C). Refer to the Appendix for the full model. The current consists of two components: a weak input $S_{i,l}^E(t)$ from the optimizable dynamics and a strong input from the balance dynamics $B_{i,l}^E(t)$:

$$I_{i,l}^E(t) = S_{i,l}^E(t) + B_{i,l}^E(t). \tag{2}$$

The **weak** input $S_{i,l}^E(t)$ is composed of three components: the excitatory feedforward current from the previous layer $\Gamma_{i,l-1}^X(t)$, the excitatory recurrent current from the same layer $\Gamma_{i,l}^E(t)$, and the inhibitory recurrent current from the same layer $\Gamma_{i,l}^{I_d}(t)$,

$$S_{i,l}^E(t) = \Gamma_{i,l-1}^X(t) + \Gamma_{i,l}^E(t) + \Gamma_{i,l}^{I_d}(t). \tag{3}$$

Each component of the weak input $S_{i,l}^E(t)$ is mediated by a set of weak and trainable connection weight variables $w$,

$$\Gamma_{i,\cdot}^b(t) = \sum_j w_{i,j}^{Eb} c_j^b(t), \tag{4}$$

where $b = X, E, I_d$. $w_{i,j}^{\mathrm{E}b}$ denotes the synaptic strength from neuron $j$ of population $b$ in the corresponding layer to neuron $i$ in the $E$ population. Importantly, we let $w_{i,j}^{\mathrm{E}b}$ scale with $1/K$ ($K$ being the number of connections). $c_j^b(t)$ represents the synaptic current input from neuron $j$ of population $b$ and is given by,

$$c_j^b(t) = \sum_k \frac{1}{\tau^s} e^{-(t-t_{j,k})/\tau^s}, \quad b = X, E, I_\mathrm{d}, I_\mathrm{p}. \tag{5}$$

where $t_{j,k}$ denotes the spike time of the $k$th spike of neuron $j$, and $\tau_{\mathrm{fast}}^s$ is the synaptic time constant.

The **strong** input $B_{i,l}^{\mathrm{E}}(t)$ from the balanced dynamics is composed of four components: the excitatory feedforward input current from the previous layer $\Omega_{i,l-1}^X(t)$, the excitatory recurrent current from the same layer $\Omega_{i,l}^E(t)$, the inhibitory recurrent current from the same layer $\Omega_{i,l}^{I_p}(t)$, and a shunting inhibition term $\mathrm{SI}_{i,l}^{I_p}(t)$ resulted from the on-path inhibition effect of PV neurons,

$$B_{i,l}^{\mathrm{E}}(t) = \Omega_{i,l-1}^X(t) + \Omega_{i,l}^E(t) + \Omega_{i,l}^{I_p}(t) + \mathrm{SI}_{i,l}^{I_p}(t). \tag{6}$$

Components of the strong input $B_{i,l}^{\mathrm{E}}(t)$ are mediated by a set of strong and untrainable connection weight variables $g$,

$$\Omega_{i,\cdot}^b(t) = \sum_j p_{i,j} g_{i,j}^{\mathrm{E}b} c_j^b(t), \tag{7}$$

where $b = X, E, I_p$, $p_{i,j} = \{1, 0\}$ denotes that neurons $i$ and $j$ are connected or unconnected, respectively. $g_{i,j}^{\mathrm{E}b}$ denotes the synaptic strength from neuron $j$ of population $b$ in the corresponding layer to neuron $i$ in the $E$ population. Importantly, we let $g_{i,j}^{\mathrm{E}b}$ scale with $1/\sqrt{N}$. We also add a shunting inhibition term $SI_{i,l}(t)$ to account for the on-path effect from the perisomatic inhibition of PV neurons (Isaacson & Scanziani, 2011), with dynamics given by (Hao et al., 2009),

$$\mathrm{SI}_{i,l}(t) = \kappa I_{i,l}^{\mathrm{EPSC}} \Omega^{\mathrm{I_p}}, \tag{8}$$

with $\kappa$ denotes the shunting inhibition strength, and $I_{i,l}^{\mathrm{EPSCs}}$ denotes the total EPSCs received by the neuron, *i.e.*, the total positive inputs. Without the shunting inhibition term, equation 6 conforms with classical E-I balance networks (van Vreeswijk & Sompolinsky, 1996). Adding the shunting inhibition term does not significantly alter the dynamics, while helping to achieve the so-called detailed-balanced state (Xue et al., 2014) by providing inhibition proportional to excitatory input.

For the first layer, we model the excitatory feedforward input as currents directly act on the neural populations, without any synaptic dynamics. This practice makes it easier to run simulations with different time step $dt$s. For the readout layer, we use non-leaky integrator neurons without spiking dynamics. All the experiments are run with BrainPy (Wang et al., 2021; 2022) which is a general-purpose brain dynamics programming framework and provides functionalities to train spiking neural networks using surrogate gradients (Neftci et al., 2019).

## 3 MAIN RESULTS

### 3.1 BALANCE DYNAMICS AS NORMALIZATION OPERATION

Normalization operations play a pivotal role in the effective training of deep neural networks. By stabilizing variable distributions, exemplified by methods like `BatchNorm` (Ba et al., 2016), these operations address the challenge of internal covariate shift – the fluctuations in variable distributions caused by weight updates that can impede later parts of the network from converging or learning effectively. However, nearly all of these normalization techniques raise biological implausibility concerns, particularly with regard to their potential to alter input variable signs, thus violating Dale's principle. In this section, we illustrate that E-I balance dynamics mediated by strong synapses is computationally similar to normalization operations while being biologically feasible.

Let's begin by considering the scenario without the influence of weak synapses. Within the E-I balance dynamics, the significant EPSCs and IPSCs, typically of the order $\mathcal{O}(\sqrt{N})$, effectively counteract each other, resulting in a relatively modest net input, typically of the order $\mathcal{O}(1)$ (van Vreeswijk & Sompolinsky, 1998). This net input essentially serves as a background signal (as depicted in Fig.

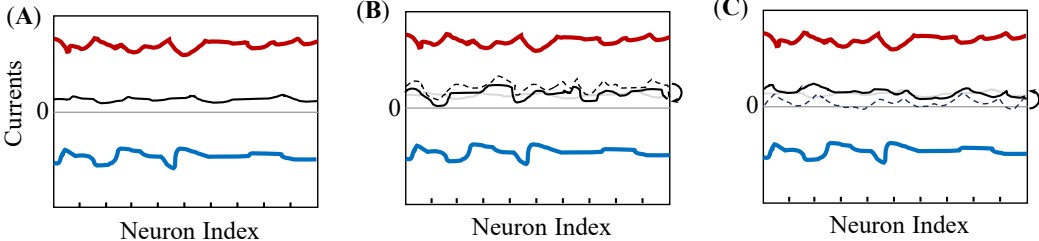

Figure 2: E-I balance dynamics effectively normalizes network activity. **(A)** Schematic representation of input currents governed by the strong E-I balance dynamics. The red and blue lines depict the combined EPSC and IPSC contributions, while the gray line represents the net input. In cases where the weak dynamics are primarily excitatory **(B)** or inhibitory **(C)**, the E-I balance dynamics recruit additional inhibition or excitation, respectively. The dashed line illustrates the hypothetical summation of net input and the current from the weak dynamics without considering feedback from the balance dynamics. The black solid line represents the actual net input current.

2A). However, the introduction of weak synapses adds an extra layer of currents superimposed upon this background signal . If these additional currents primarily consist of excitatory components, the E-I balance dynamics respond by recruiting additional inhibition to counterbalance the additional excitation from the weak dynamics (as shown in Fig. 2B), and vice versa (as indicated in Fig. 2C). Because the strong synapses within the E-I balance dynamics exhibit globally unstructured random connectivity, their impact on the network is uniform. Consequently, the E-I balance dynamics essentially provide a form of global excitation or global inhibition, contingent upon the polarity of the weak dynamics. In effect, this process functions akin to normalization operators.

The effect of the E-I balance dynamics also aligns with experimental observations indicating that the statistics of spike trains show certain similarities when different stimuli are presented, as previously documented (Fiser et al., 2004). However, the specific characteristics of neural activity can vary depending on the presence and nature of stimuli. This leads to both shared and distinct patterns in cortical dynamics, allowing for the execution of different neural computations.

To demonstrate the effectiveness of E-I balance dynamics as a normalization operation, we conducted training using two 7-layer Daleian Spiking Neural Networks (DSNNs), adhering to Dale's principle with separate excitatory and inhibitory groups, for the Fashion-MNIST recognition task (Xiao et al., 2017). One SNN incorporated trainable weak and fixed strong synapses (referred to as WS-DSNN hereafter), while the other featured fully trainable synapses initialized with `KaimingNormal` weights, following the conventional SNN setup (referred to as DSNN hereafter). To assess the stability of neuron firing rate distributions throughout training, we calculated the symmetric Kullback–Leibler (KL) divergence (Kullback & Leibler, 1951), denoted as $D_{\mathrm{KL}}^{\mathrm{sym}}$, between the distributions at initial and subsequent training steps. The symmetric KL divergence is defined as follows:

$$D_{\mathrm{KL}}^{\mathrm{sym}} = D_{\mathrm{KL}}(P\|Q) + D_{\mathrm{KL}}(Q\|P) \tag{9}$$

where $D_{\mathrm{KL}}(.\|.)$ represents the Kullback–Leibler divergence. As demonstrated in Fig. 3A, our WS-DSNN model maintains small $D_{\mathrm{KL}}^{\mathrm{sym}}$ values during training, indicating that the firing rate distribution remains relatively stable throughout the training process. In contrast, the DSNN model exhibits a significant increase in $D_{\mathrm{KL}}^{\mathrm{sym}}$ during training, signifying substantial shifts in the firing rate distribution. This outcome suggests that the issue of internal covariate shift is effectively mitigated by the E-I balance dynamics in our model, whereas a DSNN experiences notable distribution shifts. Consequently, the balanced model demonstrates faster convergence, as depicted in Fig. 3B.

## 3.2 WEAK SYNAPSES REMAIN WEAK AFTER TRAINING

During training, we do not impose any form of constraints or regularization to enforce the weak synapses to remain weak. However, interestingly, we found that these synapses maintain their small values, as illustrated in Fig. 4A. The currents produced from these weak synapses thus stay small during the entire training phase, as evidenced by Fig. 4B. Given that these weak currents do not exert

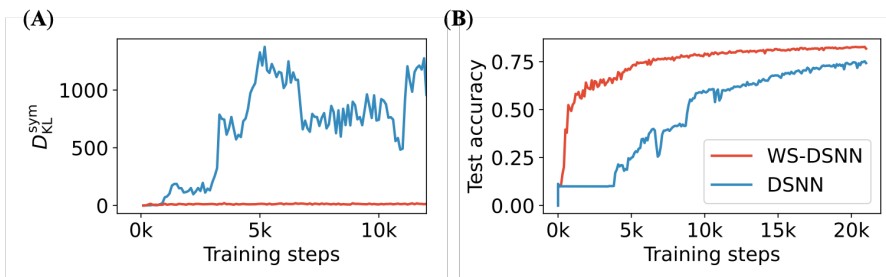

Figure 3: The E-I balance dynamics play a pivotal role in expediting training by effectively alleviating the internal covariate shift problem. (**A**) Throughout the training process, our model consistently maintains a small symmetric KL divergence $D_{\mathrm{KL}}^{\mathrm{sym}}$, while the DSNN model experiences a notable increase during training. (**B**) Our model exhibits accelerated convergence in comparison to the DSNN model when evaluated on the Fashion-MNIST dataset.

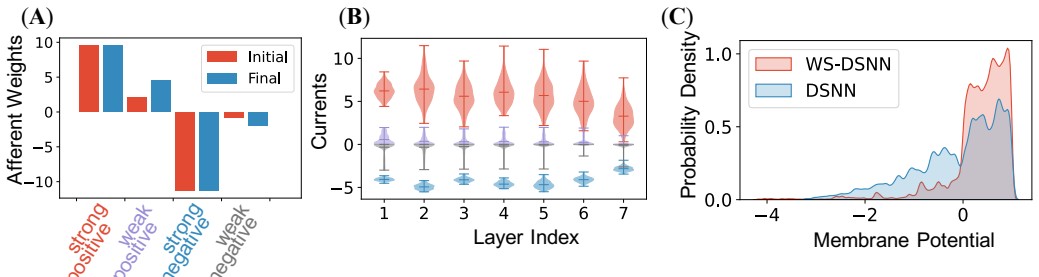

Figure 4: The weak synapses stay relatively weak after training. (**A**) A chart plot illustrating the mean afferent synapse weights of excitatory neurons before and after training. (**B**) A violin plot of the current components. The associated current component originating from weak synapses remain small during training. Colors correspond to the x-axis labels in (A). (**C**) The balanced WS-DSNN model exhibits a membrane potential distribution that is distinctly predisposed toward firing.

substantial influence on the overall magnitude of input currents, the excitatory-to-inhibitory (E/I) ratio exhibits a near-constant behavior during training, and the balance condition is thus maintained.

We attribute this phenomenon to E-I balance dynamics. E-I balance dynamics has the ability to propel neurons into a state ready for firing (Brunel, 2000; Huang et al., 2011; Tian et al., 2020). In this state, the presence of a weak dynamics component can exert a remarkably effective influence on the system's behavior. Fig. 4C offers a visual comparison of the membrane potential distributions between the balanced WS-DSNN network and a conventional Daleian network (DSNN). Notably, the balanced network exhibits a membrane potential distribution that is distinctly predisposed toward firing. Consequently, even minor adjustments to a synapse within the balanced network result in substantial changes to the network's overall state. This heightened sensitivity to synaptic changes alleviates the need for the network to apply extensive, large-scale synaptic updates.

## 3.3 TRANSITION FROM LAZY LEARNING TO RICH LEARNING

Theoretical investigations (Chizat et al., 2020; Woodworth et al., 2020) have illuminated that the scale of network initialization plays a pivotal role in determining the mode of learning within the network. A large initialization scale aligns with the "lazy" learning regime, where the part of the network before the readout layer acts akin to a fixed kernel and mostly readout weights are trained. Conversely, a small initialization scale corresponds to the "rich" learning regime, where intermediate layer weights also undergo significant changes and neural representations are thus optimized.

In an ideal scenario, the network should maintain a consistent internal neural representation even when faced with novel tasks, as this shared neural representation could potentially serve multiple

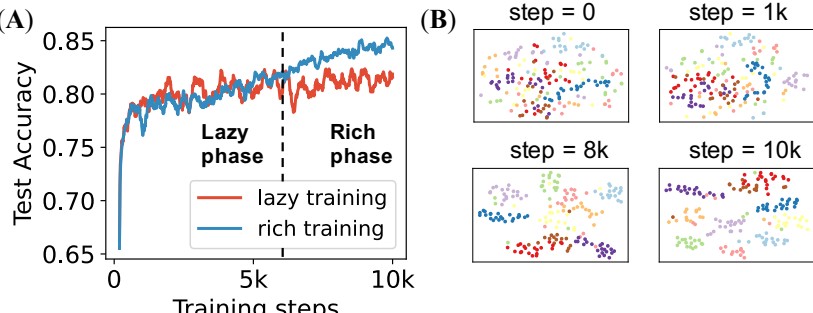

Figure 5: Transitioning from "lazy" to "rich" learning regimes on the MNIST dataset. (**A**) Networks subjected to "lazy" training (where only readout weights are updated) and "rich" training (involving updates to both readout and weak weights) exhibit a similar rapid increase in test accuracy during the initial "lazy" phase. However, networks with "rich" training can sustain performance improvements by fine-tuning weak synapses, consequently reshaping internal representations. (**B**) t-SNE visualization of network states at different training steps. During the "lazy" phase, neural representations already exhibit some degree of clustering. This initial clustering allows the network to discover a set of linear weights that yield reasonably good performance within a few training steps. With continued exposure to the same tasks, neural representations undergo alterations, resulting in enhanced clustering that aligns more closely with the specific task requirements.

purposes. However, to excel in a particular task, neural representations must adapt to the specific requirements of that task. A potential resolution to this challenge lies in the design of a network architecture capable of transitioning from "lazy" to "rich" learning. Initially, when encountering a task for the first few times, the network employs a form of "lazy" learning, swiftly training a set of readout weights to execute the task. As the network becomes more deeply engaged, it signals the significance of the current task, leading to subsequent changes in neural representation through a process reminiscent to "rich" learning.

Our findings reveal that the combination of weak and strong synapses effectively empowers the network to achieve this transition. The presence of fixed, strong weights in our model endows the network with the capability for "lazy" learning during the initial training phase (Fig. 5A). This facilitates swift adjustments in readout weights, leading to rapid improvements in training accuracy in few training steps. As the training process unfolds, we observe alterations in the neural representation of stimuli, driven by the gradual influence of updates to the weaker synapses (Fig. 5B), subsequently yielding improved performance.

## 4 EXPERIMENTS

To demonstrate the computational effectiveness of the weak-and-strong synapse architecture, we evaluate the model's performance on two widely used benchmark datasets: MNIST and Fashion-MNIST. Our primary focus lies in assessing the model's training speed and performance relative to three alternative model configurations, as outlined in Table 1. To ensure a fair comparison, we employ a consistent 7-layer architecture across all models, each layer consisting of 200 hidden units. We use the cross entropy loss as our loss function. The learning rates are set to $1 \times 10^{-3}$ for the hidden layers and $1 \times 10^{-1}$ for the readout layers, which strike an optimal balance, facilitating rapid training without compromising overall performance. The performance are shown in Table 2.

Our WS-DSNN architecture exhibited encouraging performance on both datasets, with testing accuracies either matching or surpassing those of a standard SNN equipped with `batchnorm` operations. Furthermore, WS-DSNN demonstrated training speeds comparable to regular SNNs with `batchnorm`, while the vanilla DSNN exhibited significantly slower convergence, particularly on the Fashion-MNIST dataset. However, it is noteworthy that `batchnorm` operations raise biological implausibility concerns for aforementioned reasons. Our experiments suggest that the E-I balance

Table 1: Model comparison.

| Model Type | Notes |
|---|---|
| Weak & Strong Daleian SNN (WS-DSNN) | Daleian SNN with weak and strong weights |
| Daleian SNN (DSNN) | Daleian SNN with fully trainable weights |
| Regular SNN (SNN) | `KaimingNormal` initialization |
| Regular SNN with BN (SNN-BN) | `BatchNorm` applied to currents at each layer |

Table 2: Experiment results on MNIST and Fashion-MNIST.

| | | | | |
|---|---|---|---|---|
| WS-DSNN | 0.9737 | | WS-DSNN | 0.8703 |
| DSNN | 0.9641 | | DSNN | 0.8374 |
| SNN | Failed to converge | | SNN | Failed to converge |
| SNN-BN | 0.9704 | | SNN-BN | 0.8563 |

(a) MNIST classification accuracy      (b) Fashion-MNIST classification accuracy

dynamics may be a mechanism that the brain exploits to fulfill computational roles analogous to those of `batchnorm`.

## 5 RELATED WORK

**Normalization operations** Normalization operations are fundamental to the success of deep learning, with operators like `BatchNorm` (Ioffe & Szegedy, 2015) and `LayerNorm` (Ba et al., 2016) serving as standard components in modern deep learning architectures. However, these operations pose a challenge in terms of biological plausibility. While some efforts have been made to address this issue, such as the proposal of neuronal intrinsic plasticity as a potential alternative in the study by Shaw et al. (2020), the exploration of biologically plausible alternatives remains relatively limited. One proposed avenue involves the consideration of homeostasis processes, such as synaptic scaling (Turrigiano & Nelson, 2004; Stellwagen & Malenka, 2006; Turrigiano, 2008), which could potentially stabilize network dynamics during learning. However, it's important to note that these homeostasis effects operate on much longer timescales than the rapid changes in stimulus dynamics encountered during learning. Consequently, they are unlikely to serve as viable biological candidates for normalization operations in the context of machine learning.

**Computational roles of E-I balance** The exploration of the neural function of E-I balance has primarily been confined to simplistic network models, typically featuring a single layer and excluding any learning mechanisms (van Vreeswijk & Sompolinsky, 1998; Vogels & Abbott, 2009; Lim & Goldman, 2014; Rubin et al., 2017; Tian et al., 2020). However, there have been notable exceptions and areas of specific interest within the realm of tight balance, as demonstrated in the works of Boerlin et al. (2013); Bourdoukan & Denève (2015); Thalmeier et al. (2016). These studies have delved into the dynamics of balanced networks, coupling fast and slow dynamics to approximate various forms of linear dynamics. In our investigation, we harness recent advancements in surrogate gradients (Neftci et al., 2019) and employ machine learning tasks as a means to explore the computational advantages associated with E-I balance.

**Lazy and rich learning** The brain's rapid adaptation to novel tasks is a puzzle. One proposed explanation suggests that the brain may employ a dimension-expansion strategy, a fundamental concept explored in reservoir computing (Maass et al., 2002; Jaeger & Haas, 2004; Legenstein & Maass, 2007) and extreme learning machines (Huang et al., 2006). In this approach, stimulus and context signals are exhaustively combined within a high-dimensional activity space, allowing for effective task performance through simple linear decoding. Notably, several brain structures have demonstrated this characteristic (Yassa & Stark, 2011; Cayco-Gajic & Silver, 2019; Lin et al., 2021). In contrast, neural representations should ideally adhere to a low-dimensional, task-specific manifold. This structural framework enables the filtration of irrelevant task-related information, thus minimizing interference between tasks (Ganguli et al., 2008; Chaudhuri et al., 2019). These two paradigms

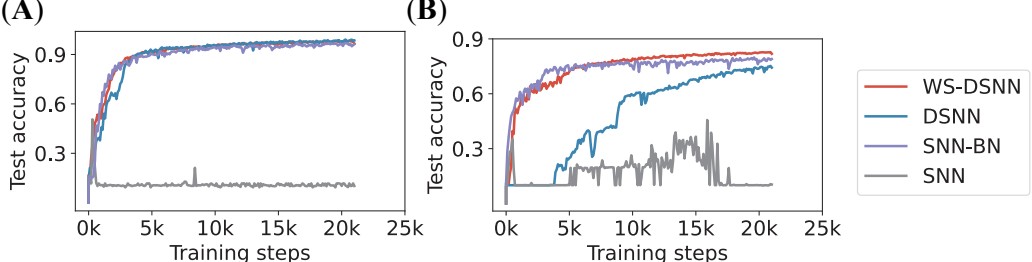

Figure 6: Training dynamics of the four models. (**A**) On the MNIST dataset, the SNN model fails to converge, while the other three models reach similar test accuracies. The difference in training speed between DSNN and WS-DSNN/SNN-BN is visible albeit marginal, which might be attributed to the dataset's simplicity. (**B**) In the context of the Fashion-MNIST dataset, the SNN model once again faces convergence challenges, while WS-DSNN and SNN-BN exhibit comparable training speeds, with WS-DSNN achieving a higher final test accuracy. DSNN encounters initial convergence difficulties due to a significant portion of deep-layer neurons remaining inactive, resulting in a gradient vanishing problem. However, it eventually converges at a slower pace.

for solving tasks (Flesch et al., 2022) have been coined as "lazy" learning (readout layer training) (Chizat et al., 2020) and "rich learning" (neural representation changes) (Woodworth et al., 2020). In the "lazy" learning regime, the network essentially emulates a random feature model (Rahimi & Recht, 2007), rendering the non-convex optimization problem effectively convex and resulting in swift convergence. Conversely, the "rich" learning regime yields highly structured representations with inductive biases, leading to more protracted convergence due to its complexity.

## 6    DISCUSSION

The balance between excitation and inhibition, considered as one of the few fundamental principles governing brain dynamics, has been extensively explored in prior research. However, reconciling the conflicting structural demands of E-I balance dynamics and neural computation requires renewed investigation. To that end, we introduce a novel E-I balance model that incorporates both fixed, strong synaptic connections and plastic, weak synapses, drawing inspiration from recent data in neuroscience experiments. This architecture not only resolves structural conflicts but also offers several computational advantages. Firstly, the balance dynamics within our model can function akin to normalization operators frequently employed in deep neural networks. This capability expedites network convergence by mitigating the internal covariate shift problem. Secondly, we observe that weak synapses tend to remain weak throughout the optimization process, thereby maintaining network balance during training. Lastly, our network exhibits a transition from "lazy" to "rich" learning, enabling it to strike a balance between learning speed and task performance.

From a biological perspective, these computational advantages hold significant implications. For instance, synaptic changes accompanying learning processes in the brain, such as the growth of new synaptic spines and modifications in existing synapses, consume both time and energy resources. If minor synaptic updates suffice for learning specific tasks, the architecture of weak-and-strong synapses could offer the brain substantial advantaging in conserving energy. Moreover, the transition from "lazy" to "rich" learning might serve as a mechanism to mitigate catastrophic forgetting.

Our findings may also shed light on the challenges associated with identifying structured weights in biological circuits during experiments. The presence of numerous unstructured neural connections may be essential for maintaining a balanced computational environment, concealing the structured connections responsible for executing cognitive tasks.

In summary, our study reveals how E-I balance dynamics can work in harmony with neural computation dynamics and facilitate the brain's learning processes. We believe that our findings provide a novel perspective on the diverse roles of E-I balance and offer valuable insights into the complex interplay between E-I balance and learning dynamics.

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

## A THE FULL NETWORK MODEL

We use the LIF-neuron model given by the equation:

$$\tau_m^b \frac{dV_{i,l}^b}{dt} = -g_L V_{i,l}^b + I_{i,l}^b, \tag{10}$$

where $b = E, I_p, I_d$ denotes the neuron type. $l$ denotes the layer index, and $i$ denotes the neuron index in that layer. $\tau_m^b$ is the membrane time constant, $g_L$ is the leaky conductance, and $I_{i,l}^b$ is the synaptic current received by the neuron.

The input currents are defined as:

$$\begin{aligned}
I_{i,l}^E(t) &= S_{i,l}^E(t) + B_{i,l}^E(t) \\
I_{i,l}^{I_p}(t) &= B_{i,l}^{I_p}(t) \\
I_{i,l}^{I_d}(t) &= S_{i,l}^{I_d}(t).
\end{aligned} \tag{11}$$

Here, $B_{i,l}^b(t)$ and $S_{i,l}^b(t)$ denote the current components from E-I balance dynamics and neural computation dynamics to neuron $i$ in population $b$ at layer $l$ at time $t$, respectively.

### A.1 E POPULATION DYNAMICS

The input current to the excitatory population consists of two components: a weak input $S_{i,l}^E(t)$ from the optimizable dynamics and a strong input from the balance dynamics $B_{i,l}^E(t)$:

$$I_{i,l}^E(t) = S_{i,l}^E(t) + B_{i,l}^E(t). \tag{12}$$

The **weak** input $S_{i,l}^E(t)$ is composed of three components: the excitatory feedforward current from the previous layer $\Gamma_{i,l-1}^X(t)$, the excitatory recurrent current from the same layer $\Gamma_{i,l}^E(t)$, and the inhibitory recurrent current from the same layer $\Gamma_{i,l}^{I_d}(t)$,

$$S_{i,l}^E(t) = \Gamma_{i,l-1}^X(t) + \Gamma_{i,l}^E(t) + \Gamma_{i,l}^{I_d}(t). \tag{13}$$

Each component of the weak input $S_{i,l}^E(t)$ is mediated by a set of weak and trainable connection weight variables $w$,

$$\Gamma_{i,.}^b(t) = \sum_j w_{i,j}^{Eb} c_j^b(t), \tag{14}$$

where $b = X, E, I_d$. $w_{i,j}^{Eb}$ denotes the synaptic strength from neuron $j$ of population $b$ in the corresponding layer to neuron $i$ in the $E$ population. Importantly, we let $w_{i,j}^{Eb}$ scale with $1/N$. $c_j^b(t)$ represents the synaptic current input from neuron $j$ of population $b$ and is given by,

$$c_j^b(t) = \sum_k \frac{1}{\tau^s} e^{-(t-t_{j,k})/\tau^s}, \quad b = X, E, I_d, I_p. \tag{15}$$

where $t_{j,k}$ denotes the spike time of the $k$th spike of neuron $j$, and $\tau_{fast}^s$ is the synaptic time constant.

The **strong** input $B_{i,l}^E(t)$ from the balanced dynamics is composed of four components: the excitatory feedforward input current from the previous layer $\Omega_{i,l-1}^X(t)$, the excitatory recurrent current from the same layer $\Omega_{i,l}^E(t)$, the inhibitory recurrent current from the same layer $\Omega_{i,l}^{I_p}(t)$, and a shunting inhibition term $SI_{i,l}^{I_p}(t)$ resulted from the on-path inhibition effect of PV neurons,

$$B_{i,l}^E(t) = \Omega_{i,l-1}^X(t) + \Omega_{i,l}^E(t) + \Omega_{i,l}^{I_p}(t) + SI_{i,l}^{I_p}(t). \tag{16}$$

Components of the strong input $B_{i,l}^E(t)$ are mediated by a set of strong and untrainable connection weight variables $g$,

$$\Omega_{i,.}^b(t) = \sum_j p_{i,j} g_{i,j}^{Eb} c_j^b(t), \tag{17}$$

Table 3: Hyperparameters

| Symbols | Values |
|---------|--------|
| $\tau_m$ | 5 |
| $g_L$ | -1 |
| $\tau_s$ | 10 |
| $\kappa$ | 0.02 |

(a) List 1

| Symbols | Values |
|---------|--------|
| $p_{\mathrm{EX}}$ | 0.1 |
| $p_{\mathrm{I_pX}}$ | 0.1 |
| $p_{\mathrm{EI_p}}$ | 0.8 |
| $p_{\mathrm{I_pI_p}}$ | 0.8 |
| $p_{\mathrm{EE}}$ | 0.1 |
| $p_{\mathrm{I_pE}}$ | 0.8 |

(b) List 2

where $b = X, E, I_p$, $p_{i,j} = \{1, 0\}$ denotes that neurons $i$ and $j$ are connected or unconnected, respectively. $g_{i,j}^{Eb}$ denotes the synaptic strength from neuron $j$ of population $b$ in the corresponding layer to neuron $i$ in the $E$ population. Importantly, we let $g_{i,j}^{Eb}$ scale with $1/\sqrt{N}$. We also add a shunting inhibition term $SI_{i,l}(t)$ to account for the on-path effect from the perisomatic inhibition of PV neurons (Isaacson & Scanziani, 2011), with dynamics given by (Hao et al., 2009),

$$\mathrm{SI}_{i,l}(t) = \kappa I_{i,l}^{\mathrm{EPSC}} \Omega^{\mathrm{I_p}}, \tag{18}$$

with $\kappa$ denotes the shunting inhibition strength, and $I_{i,l}^{\mathrm{EPSCs}}$ denotes the total EPSCs received by the neuron, *i.e.*, the total positive inputs. Without the shunting inhibition term, equation 16 conforms with classical E-I balance networks (van Vreeswijk & Sompolinsky, 1996). Adding the shunting inhibition term does not significantly alter the dynamics, while helping to achieve the so-called detailed-balanced state (Xue et al., 2014) by providing inhibition proportional to excitatory input.

### A.2 $I_p$ POPULATION DYNAMICS

The input to $I_\mathrm{p}$ population is given by:

$$I_{i,l}^{\mathrm{I_p}}(t) = B_{i,l}^{\mathrm{I_p}}(t) = \Lambda_{i,l-1}^{\mathrm{X}} + \Lambda_{i,l}^{\mathrm{E}} + \Lambda_{i,l}^{\mathrm{I_p}}. \tag{19}$$

These current components are also mediated by a set of strong and untrainable connection weight variables $g$, as in the E population,

$$\Lambda_{i,.}^{b}(t) = \sum_j p_{i,j} g_{i,j}^{Eb} c_j^b(t), \tag{20}$$

where $b = X, E, I_p$. The conductances $g_{i,j}^{\mathrm{EE}}$, $g_{i,j}^{\mathrm{EI_p}}$, $g_{i,j}^{\mathrm{I_pE}}$, and $g_{i,l}^{\mathrm{I_pI_p}}$ are the corresponding synaptic weights between neurons, and are scaled by $1/\sqrt{K}$ ($K$ being the number of connections).

### A.3 $I_d$ POPULATION DYNAMICS

The input to $I_\mathrm{d}$ population is given by:

$$I_{i,l}^{\mathrm{I_d}}(t) = S_{i,l}^{\mathrm{I_d}}(t) = \Delta_{i,l-1}^{\mathrm{X}} + \Delta_{i,l}^{\mathrm{E}} + \Delta_{i,l}^{\mathrm{I_d}}. \tag{21}$$

These current components are also mediated by a set of weak and trainable weight variables $w$, as in the E population,

$$\Delta_{i,.}^{b}(t) = \sum_j w_{i,j}^{\mathrm{I_d}b} c_j^b(t), \tag{22}$$

where $b = X, E, I_d$. $w_{i,j}^{Eb}$ are the corresponding weights between neurons and scale with $1/N$.

## B PARAMETERS

The parameters governing the E-I balance dynamics play a crucial role in our model. Specifically, we noticed that if these parameters fail to establish a balanced state, characterized by an appropriate

Table 4: Weight parameters. $K_{ab} = Np_{ab}$ denotes the number of connections from neural population $b$ to neural population $a$.

| Symbols | Initial Values |
|---------|----------------|
| $w^{\text{EX}}$ | Uniform $(0, 1/N)$ |
| $w^{\text{EE}}$ | Uniform $(0, 1/N)$ |
| $w^{\text{EI}_\text{d}}$ | Uniform $(-1/N, 0)$ |
| $w^{\text{I}_\text{d}\text{I}_\text{d}}$ | Uniform $(-1/N, 0)$ |
| $w^{\text{I}_\text{d}\text{E}}$ | Uniform $(0, 5/N)$ |

(a) Weak Synapse

| Symbols | Value |
|---------|-------|
| $g^{\text{EX}}$ | $6/\sqrt{K_{\text{EX}}}$ |
| $g^{\text{EE}}$ | $3/\sqrt{K_{\text{EE}}}$ |
| $g^{\text{EI}_\text{p}}$ | $-2.4/\sqrt{K_{\text{EI}_\text{p}}}$ |
| $g^{\text{I}_\text{p}\text{I}_\text{p}}$ | $-2/\sqrt{K_{\text{I}_\text{p}\text{I}_\text{p}}}$ |
| $g^{\text{I}_\text{p}\text{E}}$ | $0.3/\sqrt{K_{\text{I}_\text{p}\text{E}}}$ |

(b) Strong Synapse

level of excitation and inhibition, the network may encounter the vanishing gradients problem, especially in deeper layers. To address this, we first fixed a set of hyperparameters, detailed in Table 3. These hyperparameters mainly reflect a sparse connectivity within E population and dense connectivity between E and $\text{I}_\text{p}$ populations. The sparse connectivity structure within an E population is inherited from classical E-I balance models. The dense connectivity between E and $\text{I}_\text{p}$ populations is more aligned with experimental observations (Packer & Yuste, 2011; Xue et al., 2014). Then, we employ the classical E-I balance conditions outlined in prior studies like van Vreeswijk & Sompolinsky (1998) and Tian et al. (2020) as a starting point, which helps us narrow down the parameter space. Furthermore, we conduct a grid search to identify the optimal parameters for our specific tasks. During this grid search process, we observed a substantial parameter space where performance remains nearly identical. We selected one set of these parameters, which is detailed in Table 4.

