# OpenReview forum: "Fast Learning in Balanced Deep Spiking Neural Networks with Strong and Weak Synapses"
_ICLR.cc/2024/Conference — ICLR 2024 Conference Withdrawn Submission_

### Official Review · Reviewer_W5kj · 2023-10-22

**Soundness:** 2 fair
**Presentation:** 2 fair
**Contribution:** 1 poor
**Rating:** 1
**Confidence:** 5

**Summary:**

The authors make the intriguing hypothesis that inhibitory neurons in neocortical networks can be partitioned into two groups, parvalbumin-expressing interneurons (fastspiking) whose synaptic strengths are strong and fixed and responsible for excitation-inhibition balance in the cortical network, and somatostatin-expressing interneorons whose synaptic strengths are weak and can be tuned to learn particular tasks. In essence, maintaining E_I balance is delegated to one set of inhibitory neurons and learning with weak synapses to the other.

They then build a 7 layer network with all to all recurrent connections in each layer and feedforward input from the previous layer. They test this network on learning MNIST and MNIST-fashion data and report results.

**Strengths:**

The hypothesis is indeed intriguing. The authors also point to biological evidence for co-activated weak inhibitory synapses being more responsible for neural tuning than activation of strong inhibitory synapses. Finally, the introduction is very well presented.

**Weaknesses:**

The major weakness of the paper is that all results are anecdotal. They build a particular 7 layer network as described above and simply report what they see when they try to train and/or change the strength of the weak synapses from the ST population of interneurons. There is no theoretical analysis of why they see what they report. Nor have they tried to play with the network architecture to really test and try to break their own hypothesis. One immediate suggestion would be to have both populations of inhibitory neurons have a wide range of synaptic strengths, and only have the weak synapses be trainable. This clearly goes against their hypothesis that the partition of the inhibitory neurons match the strength of their synapses. Finally, why 7 layers, not 6 or 8, or any smaller number for that matter?

**Questions:**

None

---

### Official Review · Reviewer_VbNy · 2023-10-27

**Soundness:** 1 poor
**Presentation:** 1 poor
**Contribution:** 2 fair
**Rating:** 3
**Confidence:** 4

**Summary:**

This paper introduces a new approach to training spiking neural networks by keeping (pre-designated) strong weights fixed while only training the weak weights. The authors motivate this biologically by drawing comparisons to the somewhat analogous inhibitory dynamics of PV and SST cells. As the strong weights maintain EI balance in every layer, the authors claim that it effectively acts like a batch normalization procedure, thereby improving learning. Furthermore, as strong weights dominate output layer gradients in the early stages of training, the initial improvements observed are largely due to the adjustment of the output weights (lazy regime). Eventually the trainable weak weights play a bigger role in improving performance and training switches to a rich regime.

**Strengths:**

The idea of fixed and trainable weights as motivated by biology is clever. Background literature has been thoroughly reviewed and figures are clear.

**Weaknesses:**

I have one major concern (point 1) and several minor ones (points 2-5).

1. There is a lack of experiments and investigations in this paper. To the best of my understanding, in the main paper, a total of 8 (4 architectures x 2 datasets) very small networks were trained once each on MNIST or FashionMNIST-- no repeated trials to account for random seeding. All 8 networks have the same size and same number of layers. Figures 1 and 2 introduce the model, while every plot in Figures 3, 4 and 5 all come from the (single) training of the aforementioned 8 networks. There is some mention of grid searching in the last page of the supplementary, but ultimately this simply results in the strong synapses taking (a single set of) predetermined fixed values. There should minimally be some basic investigations for completeness and rigor, such as:
- experiments involving varied initializations
- experiments related to robustness against hyperparameter tuning and other factors such as learning rate (see point 2), number of layers, number of neurons (see point 4)
- experiments that investigate dynamical stability (beyond just KL from initial layer, such as steady-state behaviors)
or else the results could potentially be an artifact of lucky initialization or even simply from that specific set of pre-determined large weights.
For a paper investigating learning dynamics that uses a simple dataset like MNIST as an objective to identify interpretable learning artifacts (rich vs lazy), more rigorous experiments should be done. This also explains why there are no error bars or confidence intervals throughout the paper.

2. The authors set the readout learning rate to be 100x higher than the network learning rate, which seriously confounds any claim that lazy learning arises naturally as a result of the large untrained weights.

3. Batch normalization helps networks remain stable despite allowing weights to go beyond stable parameter regimes. On the other hand, not only does the currently proposed method not have any stability guarantee, it is training and updating weak weights safely within a dynamically-stable regime, which is the exact opposite of how batch normalization works. In other words, weights are not restricted in batch normalization, leading to faster learning with no care for stability, but are highly restricted in this method. As such, I feel that the analogy drawn is incorrect.

4. There are some arguments in the introduction and throughout the rest of the text involving the scaling of neurons. Yet, all experiments are done with the same network size. Intuitively, it is easy to understand the idea that training only weak synapses may potentially resolve the scaling contradiction, but nothing has been done to justify this point. I do acknowledge how the synapses scale by construction in the supplementary, but experiments should be done with different scalings as well.

5. I could not find what dt is, how the stimulus is actually fed into the network, and how long the network is run for, which is extremely crucial considering the continuous-time dynamics of the model. This further supports my point about the work being incomplete.

**Questions:**

See weaknesses.

---

### Official Review · Reviewer_aXEa · 2023-10-27

**Soundness:** 3 good
**Presentation:** 3 good
**Contribution:** 4 excellent
**Rating:** 6
**Confidence:** 4

**Summary:**

This paper tries to combine models of EI balance with learnable synapses to address how computations can be performed in balanced networks. It shows training benefits of including a separation between strong constant and weak learnable synapses.

**Strengths:**

This paper addresses an important and interesting open question in computational neuroscience, that of how requirements for EI balance co-exist with the needs of neural computation.

Modeling choices are reasonable and bio-inspired

Clearly written

The work provides an interesting way of interpreting biological connectivity patterns

**Weaknesses:**

This model has many details, and it is unclear which ones are crucial for its performance. It would be good to explore the roles of different features by manipulating them. For example, how crucial is the shunting inhibition? How robust are the results to different ratios between strong and weak weights? etc

It is unclear if the comparisons to other models are fair and complete. For example, the appendix notes that a wide hyperparameter search was conducted to find the hyperparameters that perform best for the WS model, but was the same done for the other models? Also it would be good to see the performance of a DSNN with BN.

The authors should cite other work that connects notions of EI balance to normalization such as https://www.sciencedirect.com/science/article/pii/S0896627314011350

It is unclear the extent to which the rich vs lazy learning results are specific to the WS model. It would be good to see the same exploration in the other models.

Figure 2 is not very visually impactful; it is hard to easily grasp the main message from it.

A little bit more of the specifics of how the weights are initialized should be included in the main text

**Questions:**

It is said that the weak weights scale with 1/K but in the appendix it says they are actually initialized with a uniform distribution with 1/N as the max. Is the 1/K in the main text a typo? Also does using a uniform distribution wherein the majority of weights are well below 1/N not cause problems for the network?

Why are there no p (probability of connection) values in the weak synapse equations? Is it because the probability is 1?

Do the g's for the strong synapse equations actually need an i,j subscript or are their values (as suggested in the appendix) identical across all pairs of cells in the given populations (and therefore specified by the superscripts)?

I am unclear on what this paragraph is trying to say: "The effect of the E-I balance dynamics also aligns with experimental observations indicating that the statistics of spike..."  It is too vague.

---

### Official Review · Reviewer_yPXB · 2023-10-30

**Soundness:** 4 excellent
**Presentation:** 4 excellent
**Contribution:** 3 good
**Rating:** 8
**Confidence:** 4

**Summary:**

The authors propose a spiking network which utilizes biologically inspired separate strong-fixed and weak-trainable weights. They show that such a network is able to maintain overall E-I balance required for basic network function, while the weaker trainable synapses are sufficient to allow training to example datasets. This paper is timed well and provides an important step to understanding the dynamics which enable biological networks to perform computation.

**Strengths:**

The paper is overall well-written and lays out rationale for most of the design choices. By combining E-I balance, an important aspect for baseline behavior of SNNs, with an additional mechanism for training the networks such that they perform functional tasks, the authors provide hope that biologically inspired networks can be trained for functional roles. Additionally, they draw parallels between their specific instantiation of E-I balance and popular modules in ANNs such as normalization. Providing such a clear link between dynamics and function is essential for communicating with non-biologists.

**Weaknesses:**

My only issue with the paper is that the MNIST And FMNIST datasets are small, with CNNs achieving ~95% accuracy, and convolutional SNNs achieving ~85% accuracy. I understand and am sympathetic to the need for smaller datasets for initial exploration, but worry that this may dissuade some of the larger ICLR community from taking interest.

**Questions:**

-	I am a bit confused regarding the weak and strong weights to/from interneurons (eg: figure 1). The rationale for strong and weak inhibition onto the excitatory cells is clear, with the parallel to SST and PV interneurons and their placement on the distal/proximal compartments. However, the excitatory populations also have separate strong and weak projections to themselves, and separate strong (proximal) and weak (distal) projections to inhibitory units. Is there any rationale for the EE, EI, and II strong vs weak weights?
-	The strong weights (eq 7) are initialized as sparse, following typical E-I network initialization, but weak weights (eq 4) are initialized as dense. Is there any biological rationale for dense weak weights, or was this a practical design choice?
-	Regarding the separate shunting term (eq 6 & 8) I had to reference back to Hao 2009 to find justification for a separate shunting term in addition to a direct inhibition from the proximal interneurons. If the authors could find room for a 0.5-1 sentence rationale it could be helpful.
-	Figure 4 shows that none of the means of the weak weights change over the course of training, but is there any enforcement (eg: constrained optimization) or evidence that none of the units switched (which would be necessary for strictly following Dale’s principle)?